# Analyzing the Driving Mechanism of Rural Transition from the Perspective of Rural–Urban Continuum: A Case Study of Suzhou, China

Yuan Yuan 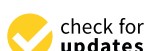, Wentao Zhao, Hongqing Li * and Han Mu

School of Public Administration, Hohai University, 8 Fochengxi Road, Jiangning District, Nanjing 211100, China; yuany@hhu.edu.cn (Y.Y.); 191314110016@hhu.edu.cn (W.Z.); 1907080105@hhu.edu.cn (H.M.)
* Correspondence: hongqing@hhu.edu.cn

**Abstract:** Rural transition has become a core topic in the study of the urban–rural relationship in China. Analyzing the transition process and sorting out the key driving factors in different periods is essential for providing critical references for the urban–rural integration and rural revitalization policy. This paper takes Suzhou, a rapidly urbanizing prefecture-level city that has experienced three obvious stages of rural transition since China's reform and opening-up, as the case area to explore the driving mechanism from the perspective of rural–urban continuum. We first construct the index system for measuring rural transition from two dimensions of rurality and urbanity. Then, we identify the core influencing factors of different phases from 1990 to 2015, employing spatial regression models and then extract the main driving mechanism. The results revealed the following key findings. (1) Rural transition in Suzhou has both proximity effects and structural effects; the development patterns of rural areas are becoming more heterogeneous. (2) From the rurality dimension, the regression coefficient of index representing grain production changes from positive to negative during the research periods, reflecting the "non-grain" trend of agricultural production in rural areas. (3) From the urbanity dimension, the regression coefficient of index promoting by foreign direct investment increases from 0.372 in 1990 to 0.829 in 2015, indicating that the external driving force of rural transition has become stronger. (4) In 2015, the regression coefficient of index representing tertiary industry reaches 0.468, meaning the modern service industry has played an increasingly important role in rural development. Our study provides valuable insights into the dynamic change of driving mechanism of rural transition at the town level, argues that the general trend of viewing transition process as rurality weakens and urbanity enhances could be replaced by multifunctional pathways. This study supplements existing research to understand new phenomena during the transition process, the latter offer implications for policy-making, such as grain security, spatial spillovers, and rural tourism.

**Keywords:** rural transition; multifunctional; rurality; "non-grain" trend; driving mechanism; spatial regression model; Suzhou



## 1. Introduction

As rapid industrialization and urbanization continue, rural transition and sustainable rural development become important issues at the forefront of research on the topic of the urban–rural relationship [1]. Along with the process of rural transition, the pulling force of cities and the pushing force of rural areas make cities and villages, and agricultural activities and non-agricultural activities, are closely linked [2]. Further, through infrastructure and information and communication technology (ICT), the differences between urban and rural blurred [3], reminding us that differences are best understood as a continuum, not a dichotomy [4]. Rather than a bounded territorial space, the rural–urban continuum views the interface between rural and urban areas as a place of exchange and socioeconomic interaction. It is a dynamic, multi-scalar settlement system that merges nodal activities

with inter-nodal flows of people, resources, and information [3]. Consequently, multiple measures have been developed expressing the gradation in rurality [5,6] or urbanity [7] to analyze issues of economic and social development. A prominent example is the rural–urban continuum codes (RUCC) produced by the U.S. Department of Agriculture beginning in the 1970s [8], which has been used extensively in contexts of analyzing variation along the rural–urban continuum of more economic and social activities [9]. Clearly, rural–urban linkages that produce the urban–rural continuum are important factors for policymakers to take into account when allocating resources or designing programs, and, thus, are crucial to analyze the driving mechanism of rural transition.

Historically, rural transition and rural areas were intrinsically associated with non-urbanization and agriculture [10], but this paradigm does not adequately describe today's complex reality [11,12]. In China, regions with higher levels of urbanization and relatively developed economies have become more heterogeneous with respect to the patterns, elements, structures, and organizational relationships of rural space [13]. Many villages have urbanized while others have evolved towards specialization, such as historical and culturally protected villages, tourism villages, industrial villages, and modern agricultural villages [14,15]. Some villages have declined or even disappeared. Rural transition in China is a dynamic, multi-scalar, and hybrid process that shares similar elements and experiences with rural restructuring as it occurred in some developed countries. However, rural transition in China is also strongly shaped by the country's distinct political, economic, social, and cultural context [16,17]. With the new millennium, both urban and rural spaces in China have entered into a new era of accelerated reconstruction against the backdrop of globalization, industrialization, and urbanization [18,19]. With the successive implementation of national policies, such as New Rural Construction and Beautiful Countryside Construction, the transformation and reconstruction of rural space has accelerated [20,21]. Moreover, in the context of the national strategy of Rural Revitalization, agriculture is increasingly serving multiple functions, such as providing food security, social stability, and ecological products, which has led to more attention being given to the study of multifunctional rurality [22].

The objective of this study is to analyze dynamic change of the driving mechanism of rural transition from a relative micro-scale, such as a specific city where urban and rural areas have transitioned from "one-way flow" to "bilateral interaction", and from "urban bias" to "urban–rural integration" [23]. Therefore, in this paper, we take Suzhou—a rapidly urbanizing prefecture-level city located in the Yangtze River Delta (YRD) region and has experienced three obvious stages of rural transition since the reform and opening up in the late 1980s—as the study area. We construct an index system of both rurality and urbanity when exploring the driving mechanism of rural transition from 1990 to 2015, and downscales the evaluation unit to the town level. Since existing research focus more on rurality evaluation from a macro-scale (i.e., national [5] and regional [2]), we tend to prove that when viewed through the perspective of the rural–urban continuum, which generally holds that rural transition occurs with the weakening of rurality and the enhancement of urbanity and tends to be heterogeneous across different regions, it becomes apparent that there could be multiple, hybrid driving forces behind these shifts [24,25].

The paper proceeds as follows. The next section provides a literature review about the rural transition and introduces the study area. Section 3 displays the index system for measuring rurality and urbanity as well as the spatial regression models employed in this study. In Section 4, we analyze the regression coefficients of indices based on the models and sort out key driving factors in different periods. The last two sections conclude by summarizing the main driving mechanism of rural transition and offering the limitation of the study and its implications.

## 2. Background and Study Area

### 2.1. Literature Review on Rural Transition

Academic research on rural transition first emerged in the 1990s. At that time, a central debate that framed the discussion on rural change has been the transition from "productivist" to "post-productivist" agricultural and rural spaces in advanced economies [26,27]. With the rapid development of economic globalization and regional integration as well as the reverse urbanization that has happened in developed countries [28], rural transition has gradually accelerated [29]. Against the backdrop of post-productivism, the boundaries between urban and rural spaces are blurring [30], and studies on the spatial evolution have shifted and extended to the functional transformation [31], the restructuring of rural space and the presence of rural spatial heterogeneity [12,32]. However, since the rise of counter-urbanization [33], rural space in developed western countries has begun to experience diversification and undergo the process of de-agrarianization [34] and gentrification [35]. Many scholars, thus, believe that the concept of multifunctional agriculture better covers the current trend of agricultural and rural development [36]. Most importantly, the concept of multifunctionality has been developed in response to public concerns about major changes taking place in agricultural and rural regions, such as a decline in the importance of agriculture in rural economics despite the fact that it still plays a crucial role [37]. Multifunctional transition is, thus, considered to be a more comprehensive trajectory that can help align post-modern agriculture with the needs of developed societies [38].

From the perspective of multifunctional rural transition, the ongoing globalization process means that agriculture is no longer the only driving force for development in the rural communities of many developed and developing countries [39], especially in Asian countries [40]. However, during the transformation of a traditional, self-sufficient agricultural society to a majority urban country [41], problems have arisen. For example, due to the acceleration of industrialization and urbanization, the land demand for construction has further increased, and a large amount of cultivated land has been converted to construction land [42]. Other problems include hollowing villages [16], non-grain production [43], and so on. To solve the problems, Korea promoted a rural modernization program—the "Saemaul Undong" (i.e., New Village Movement) [44]. Japan launched an agricultural industrial policy—the "one village, one product" strategy aiming at reversing the situation of rural talents, capital outflow, and industrial shrinkage [45]. China put forward the rural revitalization strategy plan in 2018, aims at solving key problems relating to rural transition and improving the competitiveness of sustainable development through the achievement of industrial prosperity, ecological livability, rural civilization, effective governance, and prosperous life in rural areas [2]. Villages are encouraged to pursue appropriate transition pathways according to the plan, availing of their own resource endowments or locational advantages to become revitalized in the long run [46,47]. So far, Chinese scholars have carried out systematic studies of agricultural and rural transition and development [1], most notable of these are studies on rural development and transition [48], urban–rural transformation process [49], poverty alleviation and development [50], and rural reconstruction [51].

### 2.2. Study Area

Suzhou, a prefecture-level city in Jiangsu Province, is adjacent to Shanghai (Figure 1). Almost 40 years after the reform and opening-up policy was implemented, Suzhou has had one of the highest rates of urbanization in the YRD region [52,53]. The urbanization rate in Suzhou surged from 19.44% in 1982 to 74.90% in 2015. In 2015, the per capita GDP of Suzhou reached 21,948 USD, and the per capita disposable income of urban and rural residents was 8090 USD and 4107 USD. Moreover, the urban–rural income gap of Suzhou is only 1.97, which is the smallest among all the prefecture-level cities in China. In addition, Suzhou had 71 towns and 95 *xiangs* (another administrative unit similar to towns in China) in 1990. As a result of township merging movement (*chexiangbingzhen*) during the rapid urbanization [2], all *xiangs* in Suzhou were merged into cities or towns, and there are

55 towns in 2015. Along with the process, construction land of urban area in Suzhou has expanded from less than 70 km$^2$ in 1990 to more than 700 km$^2$ in 2015, while rural area has only expanded from 600 km$^2$ to 700 km$^2$.

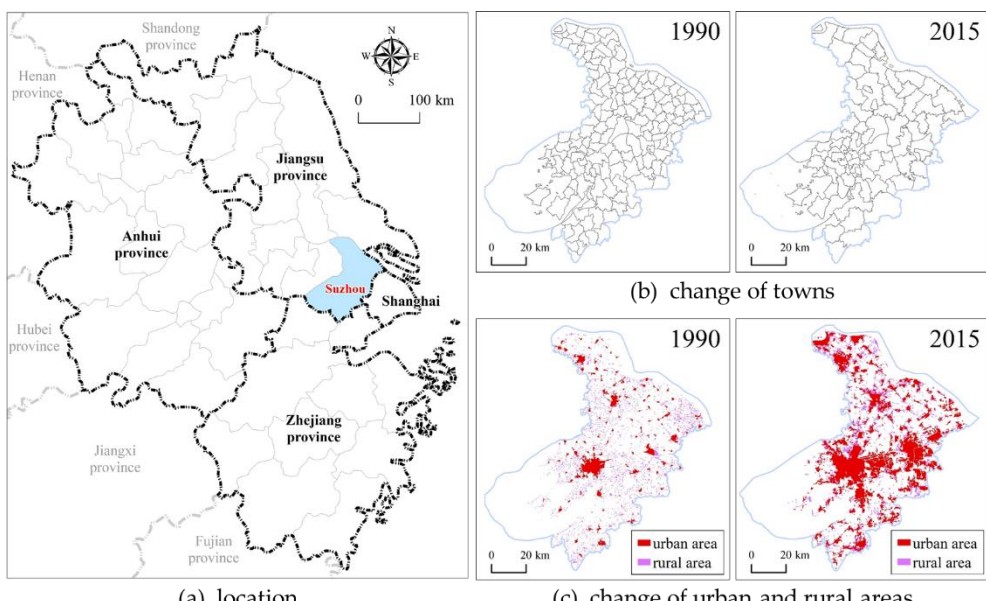

**Figure 1.** (**a**) Location of Suzhou in YRD region, China; (**b**) change of towns in Suzhou from 1990 to 2015; (**c**) change of urban and rural areas in Suzhou from 1990 to 2015.

As one of the regions in China that has experienced rapid industrialization and urbanization, Suzhou has undergone three obvious stages of rural transition. In 1980s, the development of rural industry was promoted by the Sunan model as a result of the reform and opening-up policy [54]. Meanwhile, the emerging rural industry greatly improved agricultural production efficiency, and Suzhou's rural economy has embarked on a path toward the comprehensive development of the primary and secondary industries. For the first time, Suzhou has realized the transformation of leading industrial structures by replacing agriculture with a secondary industry. In the late 1990s, rural urbanization construction began to be promoted by the new Sunan model. With the overall decline of local industry, the driving force of economic development gradually changed from "endogenous-driven" development promoted by township and village enterprises to "foreign investment-driven" development [55,56]. Particularly, the "three concentrations" strategy derived from the new Sunan model effectively guided rural industry to concentrate on industrial parks or towns, and villages tended to lean towards "simulated urbanization". The beginning of the 21st century saw a shift from urban-biased circumstances to the complete support of agriculture, villages, and peasants in China [57]. In this context, the Rural Revitalization strategy was put forward in 2017 as a coordinated urban–rural integration policy. It aims to solve the major problems of rural development and improve the competitiveness of sustainable development by realizing industrial prosperity, ecological livability, rural civilization, effective governance, and prosperity in rural areas [58,59]. The economic, ecological, and social values of the countryside have been re-recognized, and rural regions have been transformed into multifunctional post-modern spaces.

## 3. Materials and Methods

### 3.1. Indices and Data

Cloke (1986) adopted the index system of rurality to measure rural transition in different regions of the UK [60]. The index system has since become an important concept that researchers use to describe rural development, who believed that the index provides a useful tool which is able to give an insight into the process of rural change over time [5]. Zhang (1998) introduced the concept of rurality to rural geography research in China in

*Rural Spatial System and Its Evolution* [61]. Further, Zhang proposed that urban and rural regions are continuous with no breaking point between them, which could be seen as a rural–urban continuum. Areas of strong urbanity have weak rurality, which means that regions dominated by rurality can be defined as rural areas, while cities are areas of strong urbanity. The weakening of rurality and the strengthening of urbanity are the general trends of rural development in rapid urbanization areas. This interpretation is based on the analysis of China's urbanization process [62]. Since the reform and opening-up policy, the "Sunan model" in eastern coastal areas of China has dominated the development of rural industrialization and urbanization [55], which reflects the trajectory of rural development at that time, namely, the urban–rural dual linear transition.

China's large land mass and diversity have made the conceptualization of rurality a difficult task [62]. In many parts of China, "rurality" still means agriculture. However, in many villages non-agricultural activities have developed side by side with cultivation. More and more villagers have become involved in non-agricultural production, but they have not become divorced from agriculture. Scholars are increasingly questioning the enduring rural–urban dichotomy, emphasize that rural–urban interfaces no longer stand for the beginning and end of rurality and urbanity, but accommodate hybrid forms of human settlement [63]. Enlightened by the above studies and others, we tried to define the rural space from the perspective of rural–urban continuum. In short, rurality means rural characteristics while urbanity is urban characteristics in rural area. At present, the observation units of rurality in China are mainly centered on the county level and above [5], and very little research has been conducted at the town level. In 1995, Suzhou employed a management style where villages were governed by towns. At the same time, the main method of promoting integrated urban–rural development in Suzhou involves a "downward conduction of development rights", which means that development, finance, and administration matters are decided at the country level and then passed on to the town and village levels in a top-down approach. Therefore, the township-level role of Suzhou is particularly critical. The township unit is also the main body that oversees modern agricultural development, centralized residence of villagers, infrastructure construction, and ecological and cultural protection within their respective jurisdictions.

When establishing the index system of rurality and urbanity, basically, the indices must be combined with previous studies and taken into account the rural and urban characteristics in rural areas. At the same time, it is crucial that the indices can be both measured and quantified, and they must be easy to update at regular intervals [5]. Based on this context and considering the limited statistical materials and lack of consistency in the 1990–2015 indices, we constructed an index system of rurality and urbanity for the measurement of rural transition in Suzhou at the town level. The indices used in this paper came from Suzhou Statistical Yearbooks, which contain economic and social statistics of towns in Suzhou from as early as the 1980s. Moreover, considering that towns in Suzhou are different sizes, it was wiser to use efficiency indicators to measure rural transition in each town instead of employing aggregate indices, which could be affected by the different sizes of towns. This paper selected 12 indices from two dimensions of rurality and urbanity, which could largely reflect both agricultural and non-agricultural activities in rural areas, as well as agricultural and non-agricultural production. As shown in Table 1, R1 to R6 were used to measure rurality. These indices represent the quality of resource endowment and various inputs and outputs of agricultural industry in different towns. Another six indices numbered from U1 to U6 were used to measure the urbanity and determine the level of modernization and urbanization as well as the economic conditions of villagers.

**Table 1.** Index system for measuring rural transition in Suzhou.

| Indices | | Explanation |
|---|---|---|
| Number | Index and Unit | |
| R1 | Proportion of primary industry (%) | Proportion of the primary industry in the gross domestic product |
| R2 | Per capita cultivated area (mu) | Average amount of cultivated land area to year-end population |
| R3 | Per capita grain output (kg) | Average amount of grain output to year-end population |
| R4 | Power of agricultural machinery (kw/ha.) | Average amount of dynamics of farm machinery to sowing areas |
| R5 | Proportion of sown areas of non-farm crops (%) | Proportion of non-grain in the sowing area of crops |
| R6 | Per capita output of aquatic products (kg) | Ratio of output of aquatic products to year-end population |
| U1 | Proportion of secondary industry (%) | Proportion of secondary industry in the gross domestic product |
| U2 | Proportion of tertiary industry (%) | Proportion of the tertiary industry in the gross domestic product |
| U3 | Proportion of employed population (%) | Ratio of employees in TVEs to year-end population |
| U4 | Per capita pre-tax profits of TVEs (yuan) | Average amount of profits and taxes of TVEs to year-end population |
| U5 | Per capita rural electricity consumption (kwh) | Average amount of rural electricity consumption to year-end population |
| U6 | Per capita net income of villagers (yuan) | Average amount of income from rural residents |

*3.2. Methods*

Considering the obvious spatial autocorrelation between rurality and urbanity of the township-scale unit in the process of rural transition in Suzhou, spatial regression models were selected to analyze the driving factors of rural transition. Compared to the traditional regression model, the spatial regression model accounts for spatial dependence. In order to obtain the most accurate coefficient of variation from the classic ordinary least squares (OLS) model, the spatial lag model (SLM), and the spatial error model (SEM) were employed to describe and explain the related problems arising from spatial effects [64].

Considering that the urban and rural development of towns in Suzhou have taken economic growth as the key basis of socioeconomic progress since the reform and opening-up, GDP per capita, which stands for the degree of economic and social development, was used as the dependent variable, and the indicators of both rurality and urbanity index system were used as explanatory variables. The Z-score standardized was carried out. The OLS model set in this paper is as follows:

$$Y = \beta_0 + \sum_{i=1}^{m} X_i \beta_i + \varepsilon \tag{1}$$

where *i* represents different explanatory variables; *m* is the number of explanatory variables; $Y$ is the standardized result of dependent variable; $X_i$ (R1 to R6, and U1 to U6) is the standardized result of explanatory variable; $\beta_0$ is the constant coefficient; $\beta_i$ is the regression coefficient of explanatory variables; $\varepsilon$ is the random error term.

SLM mainly explores whether there was a diffusion phenomenon (proximity effect) of each variable in an area. SLM is as follows:

$$Y = \rho WY + \sum_{i=1}^{m} X_i \beta_i + \varepsilon \tag{2}$$

where $\rho$ is the spatial correlation coefficient; $W$ is the spatial weight matrix, and threshold distance is adopted in this paper; other variables are defined as above.

SEM was mainly used in the case of spatial autocorrelation of the residual term (structure effect). The model is as follows:

$$Y = \sum_{i=1}^{m} X_i \beta_i + \varepsilon \tag{3}$$

$$\varepsilon = \lambda W \varepsilon + \mu \tag{4}$$

where $\lambda$ is the spatial autocorrelation coefficient; $\mu$ is the random error vector of normal distribution; other variables are defined as above.

In addition to the goodness of fitting $R^2$ test, common test criteria include log likelihood (LogL), Akaike information criterion (AIC), and Schwartz criterion (SC). The larger the LogL is, the smaller the AIC and SC are, and the better the model fitting effect is [65]. The

indicators can be used to compare the classical linear regression model estimated by OLS, SLM, and SEM.

## 4. Results

### 4.1. Collinearity Diagnostics of Indicators

Software SPSS was used for multi-collinearity diagnosis of the standardized variables. The "entry method" is used to enter the independent variables into the model. Collinearity diagnosis provides two test results: tolerance (TOL) and variance inflation factor (VIF). When TOL is less than 0.1, there is serious collinearity. When VIF is greater than 10, there is strong multi-collinearity.

According to the test results, U1 in 1990, 2000, 2005, and 2015 and U2 in 2010 are collinear, which should be ignored in the spatial regression model for that year. Excluding collinear variables, the TOL and VIF statistics for retaining variables are shown in Table 2.

**Table 2.** Statistical results of collinearity of index system.

| Index | 1990 | | 1995 | | 2000 | | 2005 | | 2010 | | 2015 | |
|---|---|---|---|---|---|---|---|---|---|---|---|---|
| | TOL | VIF | TOL | VIF | TOL | VIF | TOL | VIF | TOL | VIF | TOL | VIF |
| R1 | 0.495 | 2.020 | 0.537 | 1.861 | 0.550 | 1.817 | 0.441 | 2.269 | 0.306 | 3.272 | 0.299 | 3.341 |
| R2 | -[1] | - | 0.204 | 4.899 | 0.342 | 2.927 | 0.264 | 3.791 | 0.105 | 9.490 | 0.255 | 3.924 |
| R3 | 0.847 | 1.181 | 0.160 | 6.238 | 0.257 | 3.890 | 0.241 | 4.153 | 0.107 | 9.341 | 0.312 | 3.208 |
| R4 | - | - | 0.455 | 2.197 | 0.502 | 1.994 | 0.692 | 1.446 | 0.429 | 2.334 | 0.496 | 2.018 |
| R5 | - | - | 0.396 | 2.525 | 0.529 | 1.890 | 0.753 | 1.327 | 0.619 | 1.615 | 0.567 | 1.762 |
| R6 | - | - | 0.429 | 2.330 | 0.443 | 2.257 | 0.859 | 1.164 | 0.500 | 2.000 | 0.796 | 1.257 |
| U1 | - | - | - | - | - | - | - | - | 0.551 | 1.814 | - | - |
| U2 | 0.569 | 1.758 | - | - | 0.733 | 1.365 | 0.696 | 1.437 | - | - | 0.595 | 1.681 |
| U3 | - | - | 0.524 | 1.909 | 0.625 | 1.600 | 0.765 | 1.307 | 0.646 | 1.547 | 0.668 | 1.496 |
| U4 | - | - | 0.624 | 1.602 | 0.487 | 2.051 | 0.510 | 1.962 | 0.124 | 8.040 | 0.659 | 1.517 |
| U5 | - | - | 0.791 | 1.264 | 0.670 | 1.491 | 0.661 | 1.514 | 0.118 | 8.447 | - | - |
| U6 | 0.804 | 1.244 | 0.689 | 1.452 | 0.698 | 1.432 | 0.456 | 2.192 | 0.458 | 2.182 | 0.641 | 1.560 |

"-" means blank. There is no such indicator in the current year.

### 4.2. Statistical Test and Comparison of Models

Software GeoDa is used to test and estimate the OLS, SLM, and SEM models. The fitting results are shown in Table 3. The goodness of fit ($R^2$) test values of SLM and SEM in each year are basically higher than those of the OLS model. When further comparing the LogL, AIC, and SC, the results show that values of LogL of SLM and SEM are higher than OLS, while values of AIC and SC of SLM and SEM are lower in each year (Table 3). It is clear that the spatial regression models are superior to the OLS, and the SEM is better than the SLM in most years.

**Table 3.** Statistical test results of model.

| Test Item | 1990 | | | 1995 | | | 2000 | | |
|---|---|---|---|---|---|---|---|---|---|
| | OLS | SLM | SEM | OLS | SLM | SEM | OLS | SLM | SEM |
| $R^2$ | 0.523 | 0.541 | 0.585 | 0.650 | 0.654 | 0.650 | 0.572 | 0.633 | 0.693 |
| LogL | −150.511 | −150.026 | −144.878 | −140.855 | −140.097 | −140.854 | −123.838 | −115.845 | −108.622 |
| AIC | 311.023 | 312.051 | 299.757 | 303.710 | 304.194 | 303.707 | 271.676 | 257.691 | 241.244 |
| SC | 325.941 | 329.953 | 314.675 | 337.399 | 340.945 | 337.396 | 305.615 | 294.459 | 275.184 |

**Table 3.** *Cont.*

| Test Item | 2005 | | | 2010 | | | 2015 | | |
|---|---|---|---|---|---|---|---|---|---|
| | **OLS** | **SLM** | **SEM** | **OLS** | **SLM** | **SEM** | **OLS** | **SLM** | **SEM** |
| $R^2$ | 0.635 | 0.640 | 0.650 | 0.370 | 0.376 | 0.388 | 0.804 | 0.806 | 0.806 |
| LogL | −59.881 | −59.501 | −59.010 | −69.588 | −69.402 | −69.263 | −31.464 | −31.305 | −31.398 |
| AIC | 143.762 | 145.002 | 142.020 | 163.175 | 164.805 | 162.526 | 84.928 | 86.610 | 84.797 |
| SC | 170.038 | 173.468 | 168.296 | 188.105 | 191.813 | 187.456 | 106.601 | 110.253 | 106.470 |

Diagnostics for spatial dependence also proves the results (Table 4). For the data of 1990, the LM-Lag test is not significant, the LM-Error test is significant at the 0.01 level (LM-Error = 7.126, $p$ = 0.008), and the SEM model is recommended. For 1995, the LM-Lag test is significant at the 0.01 level (LM-Lag = 11.068, $p$ = 0.001), the LM-Error test is not significant, and the SLM model is recommended. For the other years, the LM-Lag test and LM-Error test are significant at the 0.01 level, and the SLM and SEM model are recommended.

**Table 4.** Diagnostics for spatial dependence.

| Test Item | 1990 | | 1995 | | 2000 | |
|---|---|---|---|---|---|---|
| | **Value** | **Prob** | **Value** | **Prob** | **Value** | **Prob** |
| Lagrange Multiplier (Lag) | 0.022 | 0.883 | 11.068 | 0.001 | 18.531 | 0.000 |
| Robust LM (Lag) | 7.288 | 0.007 | 12.502 | 0.000 | 0.146 | 0.702 |
| Lagrange Multiplier (Error) | 7.126 | 0.008 | 1.527 | 0.217 | 27.419 | 0.000 |
| Robust LM (Error) | 14.393 | 0.000 | 2.961 | 0.085 | 9.034 | 0.003 |
| **Test item** | **2005** | | **2010** | | **2015** | |
| | **Value** | **Prob** | **Value** | **Prob** | **Value** | **Prob** |
| Lagrange Multiplier (lag) | 21.121 | 0.000 | 17.407 | 0.000 | 20.322 | 0.000 |
| Robust LM (lag) | 0.325 | 0.569 | 0.447 | 0.504 | 0.269 | 0.604 |
| Lagrange Multiplier (error) | 28.682 | 0.000 | 23.333 | 0.000 | 25.490 | 0.000 |
| Robust LM (error) | 7.886 | 0.005 | 6.373 | 0.012 | 5.437 | 0.020 |

*4.3. Model Estimation Results*

4.3.1. OLS Model Estimation Results

As shown in Table 5, five variables—proportion of primary industry (R1), per capita grain output (R3), proportion of secondary industry (U1), proportion of tertiary industry (U2), and per capita net income of villagers (U6) in 1990—all passed the significance test at the 1% level. Among them, the regression coefficients of R1 and U2 are negative, and the regression coefficients of R3 and U6 are positive. By 2015, only four indicators, namely, proportion of sown areas of non-farm crops (R5), proportion of tertiary industry (U2), per capita pre-tax profits of TVEs (U4), and per capita net income of villagers (U6), passed the significance test.

4.3.2. SLM and SEM Estimation Results

The estimated results of SLM and SEM are shown in Tables 6 and 7. Among all the years, only the correlation coefficient of SLM in 2000 ($\rho$ = 0.398) is significant at the 1% level. This result indicates that the rural transition of Suzhou in this particular year had a strong spatial dependence, and the proximity effect is very obvious. The estimated results of SEM show that the correlation coefficients λ in 1990, 2000, and 2005 (0.476, 0.665, and 0.364) pass the significance tests at the levels of 1% and 10%. This indicates that there is spatial heterogeneity in the years above, which is linked to the systemic differences caused by other error terms (i.e., unforeseen variables).

**Table 5.** Estimation results of OLS model (1990–2015).

| Index | OLS Model | | | | | |
|---|---|---|---|---|---|---|
| | **1990** | **1995** | **2000** | **2005** | **2010** | **2015** |
| R1 | −0.480 *** | −0.460 *** | −0.181 ** | −0.057 | −0.323 | −0.102 |
| | (0.000) | (0.000) | (0.032) | (0.647) | (0.130) | (0.418) |
| R2 | - | 0.197 * | 0.244 ** | 0.206 | 0.115 | 0.066 |
| | | (0.070) | (0.022) | (0.204) | (0.748) | (0.629) |
| R3 | 0.381 *** | 0.209 * | 0.053 | −0.029 | −0.052 | −0.090 |
| | (0.000) | (0.088) | (0.662) | (0.865) | (0.884) | (0.464) |
| R4 | - | 0.058 | 0.117 | −0.156 | 0.226 | 0.017 |
| | | (0.421) | (0.181) | (0.121) | (0.207) | (0.864) |
| R5 | - | 0.167 ** | 0.066 | 0.088 | −0.273 * | −0.154 * |
| | | (0.033) | (0.436) | (0.360) | (0.070) | (0.098) |
| R6 | - | 0.156 ** | 0.264 *** | 0.176 * | −0.095 | 0.039 |
| | | (0.038) | (0.005) | (0.052) | (0.565) | (0.616) |
| U2 | −0.230 *** | - | −0.088 | −0.004 | −0.131 | 0.470 *** |
| | (0.003) | | (0.226) | (0.971) | (0.404) | (0.000) |
| U3 | - | 0.210 *** | 0.122 | 0.165 * | −0.050 | 0.089 |
| | | (0.002) | (0.120) | (0.085) | (0.732) | (0.294) |
| U4 | - | 0.373 *** | 0.330 *** | 0.673 *** | −0.070 | 0.828*** |
| | | (0.000) | (0.000) | (0.000) | (0.831) | (0.000) |
| U5 | - | 0.188 *** | 0.214 *** | −0.050 | 0.091 | - |
| | | (0.001) | (0.005) | (0.625) | (0.787) | |
| U6 | 0.171 *** | 0.274 *** | 0.235 *** | 0.329 *** | 0.333 * | 0.184 ** |
| | (0.008) | (0.000) | (0.002) | (0.009) | (0.058) | (0.037) |

***, **, and * mean significant at the level of 1%, 5%, and 10%. U1 is omitted for collinear problem.

**Table 6.** SLM model estimation results (1990–2015).

| Index | SLM Model | | | | | |
|---|---|---|---|---|---|---|
| | **1990** | **1995** | **2000** | **2005** | **2010** | **2015** |
| R1 | −0.462 *** | −0.458 *** | −0.144 * | −0.067 | −0.354 * | −0.109 |
| | (0.000) | (0.000) | (0.052) | (0.551) | (0.060) | (0.328) |
| R2 | - | 0.190 * | 0.179 * | 0.187 | 0.120 | 0.063 |
| | | (0.067) | (0.055) | (0.197) | (0.706) | (0.600) |
| R3 | 0.367 *** | 0.193 | 0.033 | −0.005 | −0.066 | −0.086 |
| | (0.000) | (0.101) | (0.760) | (0.973) | (0.834) | (0.429) |
| R4 | - | 0.052 | 0.128 * | −0.173 * | 0.200 | 0.029 |
| | | (0.455) | (0.098) | (0.055) | (0.204) | (0.740) |
| R5 | - | 0.158 ** | 0.053 | 0.081 | −0.278 ** | −0.145 * |
| | | (0.034) | (0.479) | (0.342) | (0.034) | (0.079) |
| R6 | - | 0.162 ** | 0.213 *** | 0.187 ** | −0.070 | 0.039 |
| | | (0.023) | (0.010) | (0.019) | (0.632) | (0.568) |
| U2 | −0.222 *** | - | −0.061 | 0.002 | −0.138 | 0.463 *** |
| | (0.003) | | (0.334) | (0.982) | (0.319) | (0.000) |
| U3 | - | 0.233 *** | 0.126 * | 0.160 * | −0.059 | 0.086 |
| | | (0.000) | (0.065) | (0.060) | (0.645) | (0.246) |
| U4 | - | 0.359 *** | 0.315 *** | 0.659 *** | −0.060 | 0.825 *** |
| | | (0.000) | (0.000) | (0.000) | (0.838) | (0.000) |
| U5 | - | 0.175 *** | 0.257 *** | −0.045 | 0.098 | - |
| | | (0.001) | (0.000) | (0.621) | (0.745) | |
| U6 | 0.168 *** | 0.275 *** | 0.226 *** | 0.336 *** | 0.322 ** | 0.174 ** |
| | (0.008) | (0.000) | (0.001) | (0.002) | (0.036) | (0.023) |
| $\rho$ | 0.123 | 0.137 | 0.398 *** | −0.190 | −0.159 | 0.076 |
| | (0.284) | (0.224) | (0.000) | (0.308) | (0.494) | (0.559) |

***, **, and * mean significant at the level of 1%, 5%, and 10%. U1 is omitted for collinear problem.

**Table 7.** SEM model estimation results (1990–2015).

| Index | SEM Model | | | | | |
|---|---|---|---|---|---|---|
| | **1990** | **1995** | **2000** | **2005** | **2010** | **2015** |
| R1 | −0.550 *** | −0.460 *** | −0.180 ** | −0.069 | −0.361 ** | −0.102 |
| | (0.000) | (0.000) | (0.012) | (0.523) | (0.045) | (0.352) |
| R2 | - | 0.197 * | 0.108 | 0.189 | 0.124 | 0.090 |
| | | (0.059) | (0.279) | (0.196) | (0.691) | (0.443) |
| R3 | 0.463 *** | 0.209 * | 0.103 | −0.001 | −0.063 | −0.092 |
| | (0.000) | (0.076) | (0.362) | (0.992) | (0.834) | (0.393) |
| R4 | - | 0.056 | 0.003 | −0.147 * | 0.117 | 0.002 |
| | | (0.420) | (0.967) | (0.094) | (0.465) | (0.979) |
| R5 | - | 0.166 ** | 0.000 | 0.084 | −0.237 * | −0.145 * |
| | | (0.027) | (0.996) | (0.347) | (0.070) | (0.066) |
| R6 | - | 0.157 ** | 0.301 *** | 0.157 * | −0.021 | 0.044 |
| | | (0.029) | (0.002) | (0.060) | (0.881) | (0.505) |
| U2 | −0.172 ** | - | −0.004 | −0.008 | −0.135 | 0.468 *** |
| | (0.015) | | (0.949) | (0.927) | (0.340) | (0.000) |
| U3 | - | 0.211 *** | 0.129 ** | 0.144 * | −0.103 | 0.102 |
| | | (0.001) | (0.040) | (0.083) | (0.392) | (0.166) |
| U4 | - | 0.372 *** | 0.345 *** | 0.707 *** | −0.039 | 0.829 *** |
| | | (0.000) | (0.000) | (0.000) | (0.894) | (0.000) |
| U5 | - | 0.187 *** | 0.255 *** | −0.043 | 0.059 | - |
| | | (0.000) | (0.000) | (0.633) | (0.846) | |
| U6 | 0.229 *** | 0.274 *** | 0.285 *** | 0.333 *** | 0.342 ** | 0.201 *** |
| | (0.000) | (0.000) | (0.000) | (0.002) | (0.020) | (0.007) |
| $\lambda$ | 0.476 *** | 0.012 | 0.665 *** | 0.364 * | −0.409 | −0.207 |
| | (0.000) | (0.942) | (0.000) | (0.080) | (0.156) | (0.458) |

***, **, and * mean significant at the level of 1%, 5%, and 10%. U1 is omitted for collinear problem.

In addition to the spatial effect, the rural transition of Suzhou that took place from 1990 to 2015 was also affected by multiple characteristic variables. The estimation structures of the OLS, SLM, and SEM models are relatively robust, and the significant variables representing the models are the same, as are the positive and negative values of regression coefficients. Considering that spatial regression models are superior to the OLS (bigger $R^2$/LogL and smaller AIC/SC), analysis of the driving factors is based on the estimation results of SLM and SEM.

In case of the potential endogeneity problem, more interpretation should be addressed than that we have considered it from the angles of reverse causality and omitted variables during the establishment of index system [66], and we are aware of instrumental variable chosen from the geographical and historical perspective could deal with the endogeneity problem [67,68]. As mentioned above, the indices are combined with previous studies and take into account the rural and urban characteristics in rural areas. For the potential problem of reverse causality, many existing literatures have proved the causal links between the indices and rural economic development (the dependent variable in this paper), such as R1—proportion of primary industry [15], R3—per capita grain output, U4—per capita pre-tax profits of TVEs [54], and U5—per capita rural electricity consumption [3]. For the problem of omitted variables, the 12 indices together can reflect both agricultural and non-agricultural activities, and agricultural and non-agricultural production in rural areas, as a result of the perspective of rural–urban continuum. Likewise, we find small biases among the regression coefficients of indices from OLS, SLM and SEM models, which means estimation results of these models are robust. This could also be used to interpret the endogeneity problem.

## 5. Discussion

According to the results of the spatial regression model that look at the data from 1990 to 2015, rural transition in Suzhou has shown both proximity effect and structural

effect, which is a process full of complexity. With respect to the orientation of economic development, rural transition is not only driven by the industrial development patterns of surrounding towns and villages to form a homogeneous industrial pattern or competitive relationship, but also absorbs functional spillover from urban areas to create new economic growth points in a way that accounts for new urban–rural relations needs. In conclusion, the driving mechanism affecting rural transition in Suzhou is summarized as follows.

### 5.1. Trends of "Non-Grain" in Agricultural Production

In the early years, the two variables representing grain production capacity—per capita cultivated area (R2) and per capita grain output (R3)—show significantly positive correlations with rural economic development, and the two variables standing for non-grain production capacity (i.e., the proportion of sown areas of non-farm crops (R5) and per capita output of aquatic products (R6)). The regression coefficients of the four variables in 1995 are 0.197, 0.209, 0.166, and 0.157, respectively. The influence of the variables representing grain production capacity is stronger that represent non-grain production capacity. However, after 2000, only variable R6 is still positively significant; the regression coefficients of the variable in 2000 and 2005 are 0.301 and 0.157. This shows that the impact of "non-grain" in the process of agricultural production is increasing along with modern agriculture development in Suzhou, especially with respect to the fishery industry and the resource endowment for intertwined river networks. The total output value of fishery in 1990 is lower than that of animal husbandry, and the ratio to the total output value of the planting industry is only 2.70:1. In 2015, the total output value of fishery is 5.00 times and 3.34 times that of forestry and animal husbandry, respectively, and the ratio to the total output value of planting industry has also narrowed to 1.33:1 (Table 8).

**Table 8.** Changes in output value and structure of primary industry in Suzhou (1990–2015).

| Year | Agriculture | | Forestry | | Animal Husbandry | | Fishery | |
|---|---|---|---|---|---|---|---|---|
| | Output Value ($10^5$ Yuan) | Ratio(%) | Output Value ($10^5$ Yuan) | Ratio(%) | Output Value ($10^5$ Yuan) | Ratio(%) | Output Value ($10^5$ Yuan) | Ratio(%) |
| 1990 | 349,554 | 60.14 | 5025 | 0.86 | 119,400 | 20.54 | 107,243 | 18.45 |
| 1995 | 918,417 | 64.85 | 9834 | 0.69 | 233,110 | 16.46 | 254,812 | 17.99 |
| 2000 | 1,001,634 | 59.16 | 11,740 | 0.69 | 231,251 | 13.66 | 448,357 | 26.48 |
| 2005 | 577,417 | 39.76 | 63,554 | 4.38 | 217,068 | 14.95 | 594,059 | 40.91 |
| 2010 | 1,052,848 | 43.40 | 161,828 | 6.67 | 357,452 | 14.73 | 853,943 | 35.20 |
| 2015 | 1,728,527 | 47.12 | 258,763 | 7.05 | 387,641 | 10.57 | 1,293,042 | 35.25 |

Suzhou is a well-known water area of the Yangtze River, and fishery has become a new engine for rural economic development in the region. For example, the "clear water hairy crab" that is caught in the town of Yangchenghu recently became short in supply, and the sales channel for this particular product has been bound to the online market. Obviously, this pond fish farming industry has a continuous and significant neighborhood effect in rural areas where the densities of river net are high, and potential risks of invasive fish [69] and eutrophication [70]. However, this rapid expansion of non-grain production on cultivated land is of increasing concern regarding grain security in China [71], and apparently contradicts the national policy such as basic farmland and standardized farmland [43]. This finding on driving force of non-grain production during rural transition warn us the importance of protecting "rice bowls", as it is likely to contain more than rice in future [72].

### 5.2. External and Internal Forces of Economic Development

The four characteristic variables of urbanity (i.e., proportion of employed population (U3), per capita pre-tax profits of TVEs (U4), per capita rural electricity consumption (U5), and per capita net income of rural (U6)) show a strong positive correlation with the rural transition in Suzhou. In particular, variable U4, which reflects the external economic driving force, has a regression coefficient of 0.372, 0.345, 0.707, and 0.829 in 1990, 1995, 2000, and

2015, respectively. This result shows that the impact intensity of U4 increases year by year and is the most influential variable in the current year.

Since the reform and opening-up policy was implemented, township enterprises, as the main driving force of rural industrialization, have promoted the urbanization development of rural space in Suzhou by means of intensive, large-scale, and standardized operation of construction land and industrial parks in towns and villages. In the beginning, there were integrated development platforms for village–village alliances in towns, and then platforms for development of single village or between joint villages that were upgraded to the higher-level coordination of towns and counties. Following the early "Sunan model", the towns of Yushan and Yangshe in Suzhou have become two of the top 100 economically developed towns in China. Variable U4 highlights the external motivation of pursuing rapid economic growth based on government investment and foreign capital. Undoubtedly, improvements in transportation and ICT promoted this happen [21]. Being the nearby hinterlands in the YRD region cored by Shanghai, towns and counties in Suzhou having been beneficial from increasing spatial spillovers effect [73], reflecting the rural–urban regional interdependencies and spillovers within an urban center's "area of influence" [3].

Another variable that reflects economic development is U6, and regression coefficient also shows a stable positive correlation, which represents the endogenous motivation of rural transition. The regression coefficient of each year is 0.229, 0.274, 0.285, 0.333, 0.342, and 0.201, respectively, and the impact intensity is second only to U4. In 2020, the ratio between per capita disposable income of the urban and rural residents in Suzhou reduces to 1.89 (Figure 2), making Suzhou one of the cities with the smallest urban–rural income gaps in China. Endogenous development is an important part of sustainable rural development, and it could improve the vitality of rural communities, realize the goal of rational flow of elements between urban and rural areas or different regions, and lead to the optimal allocation of services. It could also promote the development of multiple functions and values in rural regions and the integration of industrial development and local environments, resources, society, and culture.

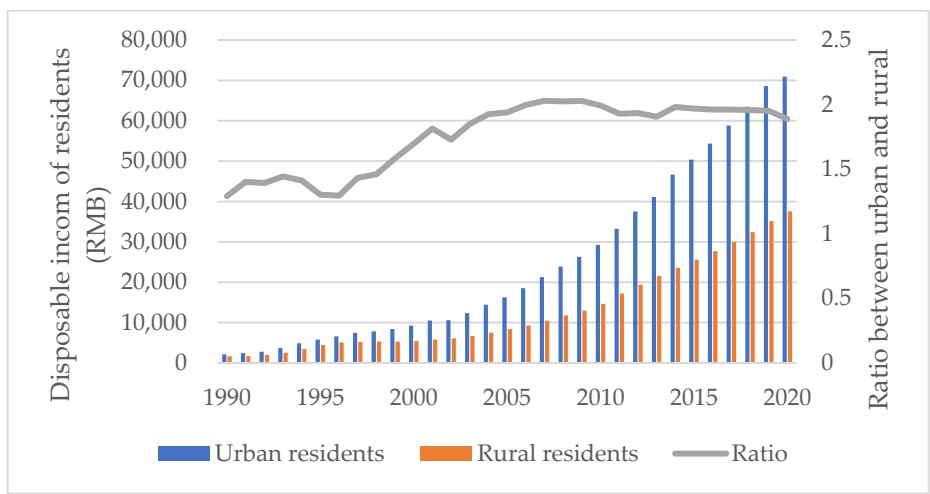

**Figure 2.** Income changes of urban and rural residents in Suzhou (1990–2020).

### 5.3. Increasing Contributions from New Industry

The estimation results of all characteristic variables from 1990 to 2015 indicate changes among different years. In 1990 and 1995, the variables of rurality and urbanity both have a significant effect on rural transition. However, the impact of the characteristic variables representing rurality has decreased since then. After 2000, this impact is mainly concentrated in the variables standing for urbanity. In 2015, the variables representing rurality have basically not passed the significance test, while most of the variables representing urbanity are still significant at the 1% level.

This change is also reflected in the estimation results of variables representing industrial proportion, such as proportion of primary industry (R1), proportion of primary industry (U1), and proportion of tertiary industry (U2). The regression coefficients of U1 in 1990 and 2010 were $-0.550$ and $-0.361$, respectively. Although the value increases, it is still negative. The reason for this is that the proportion of primary industry has decreased from 17.31% to 1.53% from 1990 to 2010, which is far less than that of the secondary and tertiary industries, though the output value of primary industry has increased from CNY 35 billion in 1990 to CNY 140.80 billion in 2010. The regression coefficient of U2 changes from $-0.172$ in 1990 to 0.468 in 2015, which represents a shift towards a positive correlation with per capital re-tax profits of TVEs (U4), which was 0.829 in 2015. There are two reasons for this result. First, the proportion of the tertiary industry in Suzhou exceeded that of the secondary industry for the first time in the same year, reaching 49.94%. Second, Suzhou, as an economically developed city, is experiencing a post-productivism transition, which has led to a reconstruction of multiple values in its rural space. Especially, the construction of beautiful countryside strategy is to allow the exploration of special resources around big cities [74]. Fresh air, leisure life, and rural life are known to attract urban tourists [2], which contributes to villagers' income through the linkage and new industry of agriculture and tourism.

## 6. Conclusions

From productivism to post-productivism, an important change has occurred in the core characteristics of agriculture [26,46]. The former theory is manifested in intensiveness, concentration, and simplification, and the latter shows more extensionality, dispersion, and diversity. Since the reform and opening up took place in the 1980s, China has been transiting from a traditional society to a modern one. Rural industrialization came first and triggered the transition process in rural spaces. Therefore, on the macro scale, industrialization and urbanization have always been regarded as the basic path for rural areas to realize the transition from traditional society to modern civilization [5,43]. While analyzing the evolution process of rural-space systems in southern Jiangsu, Zhang (1998) proposed that "under the trend of urbanization, the country's function is continually changing, and rural space is not only for the agricultural economy activity but a multi-functional space for the overall development of primary, secondary and tertiary industries" [61]. Results from spatial regression models focusing on micro-level observations (town unit in this paper) also imply that rapid urbanization can lead to multiple driving mechanisms and differentiation of rural space in Suzhou, which emphasizes the importance of spatial unit in the study of rural transition. Thus, the contribution of this paper is to argue that the general trend of viewing rural development as a process that weakens rurality and enhances urbanity can only be emphasized when approached at the macro-scale level; however, from a micro-scale level, there are multifunctional pathways of rural transition [55]. Considering the representativeness of Suzhou both at home and aboard, this conclusion applies not only to rural areas undergoing transition in China but also to villages all over the world, especially those in developing countries.

Although the perspective of rural–urban continuum has revealed dynamic change of driving mechanism of rural transition in different periods, this paper cannot analyze all types of rural areas experiencing transition in China. In this case, we only take Suzhou—a rapidly urbanizing city in the YRD region since the reform and opening up—as the typical study area. This consideration is mainly based on our evaluation unit at the town level. Compared to the existing research focus more on rurality evaluation at the county level and above [5], two major problems have limited our data collection. First, town-level data used in this paper is only provided in statistical yearbooks of prefecture-level city and below, therefore indicators in yearbooks of different cities or counties are not the same, which is difficult for constructing a universal index system. Second, the incontinuity of town-level data occurs when trace information to the earlier years, so rare cases such as Suzhou provides consistent annual statistical data of towns since the 1990s. Considering these

shortcomings, further research on comparison between Suzhou and cities with sufficient data in YRD region or other urban agglomeration experiencing rural transition in China is worth exploring.

Despite the imperfect analysis, this paper offers theoretical and practical implications for policy-making. Pathways during rural transition from the perspective of rural–urban continuum are heterogeneous across different regions because of various factors related to natural resources, cultural elements, and financial positions. For this reason, multidimensional and hybrid development pathways in which questions about the "right" and "wrong" development trajectories are increasingly difficult to answer. In this way, theoretically, a singular, absolute developmental trajectory of "modernization" does not apply to rural transition in China. This case study of Suzhou indicates that the establishment of multifunctional rural space has provided a comparative advantage that could meet the needs of pluralistic values (e.g., production and consumption, and development and protection) during the process of rural transition. Without a question, multifunctionality will become the main trend of rural development, especially in rapidly urbanized areas with location advantage and resource endowment. Considering that the modern service industry, rural tourism, and other new industries are now playing more important roles in the rural transition, encouraging the development of the tertiary industry is thus worth putting into practice in the formulation of policy for sustainable development of rural space.

**Author Contributions:** Y.Y.: Conceptualization; methodology; investigation; data curation; writing—original draft preparation; visualization; funding acquisition. W.Z.: Validation; formal analysis; writing—review and editing. H.L.: resources; supervision. H.M.: software. All authors have read and agreed to the published version of the manuscript.

**Funding:** This research was funded by the National Natural Science Foundation of China (Grants No. 42001196), the Fundamental Research Funds for the Central Universities (Grants No. B220201071).

**Institutional Review Board Statement:** Not applicable.

**Informed Consent Statement:** Not applicable.

**Conflicts of Interest:** The authors declare no conflict of interest.

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
