# Peer review of "Analyzing the Driving Mechanism of Rural Transition from the Perspective of Rural–Urban Continuum: A Case Study of Suzhou, China"

_land, doi:10.3390/land11081146_

Round 1
Reviewer 1 Report
The selected study area only covers Suzhou. With Suzhou as the only sample of rapidly urbanizing regions, the econometric model only explains the local variations in Suzhou. It is unclear what lessons from this research can be generalized to represent rapidly urbanizing regions in general. The use of a larger dataset covering all the rapidly urbanizing regions in China can help to address this concern.
The choice of econometric models needs to be justified. The three econometric models have different assumptions about the underlying data generation process. A more detailed discussion about model selection between OLS, SLM and SEM is necessary. The exclusion of other more general spatial econometric models also needs to be discussed. In addition, in Table 5 and 6, SLM seems to be more appropriate in some periods while SEM seem more appropriate in other periods. This time inconsistency in the underlying data generation process needs to be justified.
The authors describe the merge of Xiangs over time. This boundary change invalidates the reported comparisons over time. One alternative is to construct a time-consistent administrative boundary and then conduct a panel analysis.
In the model specification, the potential endogeneity problem needs to be carefully addressed.
Author Response
- The selected study area only covers Suzhou. With Suzhou as the only sample of rapidly urbanizing regions, the econometric model only explains the local variations in Suzhou. It is unclear what lessons from this research can be generalized to represent rapidly urbanizing regions in general. The use of a larger dataset covering all the rapidly urbanizing regions in China can help to address this concern.
Response: Thank you for your comments and suggestions. In the conclusions part in the revised manuscript, we admit that the deficiency of this paper is only take Suzhou as the study area. However, reasons are given to explain our consideration. In short, our analysis employing spatial regression models is based on town-level unit, two major problems have limited the data collection, namely, inconsistent indicators in yearbooks of different cities, and incontinuity of town-level data when trace to the earlier years. In this case, rare study area like Suzhou could provide consistent annual statistical data of towns since the 1990s at town level. For future study, we wish to compare Suzhou with other cities experiencing rural transition in China. Please find more details in line 513-527 (all markup mode of the word) in the revised manuscript.
In addition, we have also interpreted the necessity to analyze dynamic change of the driving mechanism from a relative micro-scale, such as a specific city, considering that too many existing research are focus on rurality evaluation from a macro-scale. Has experienced three obvious stages of rural transition since the reform and opening up, Suzhou is a typical study area for this paper. Please find this expression in line 113-121 (all markup mode of the word) in the revised introduction part.
- The choice of econometric models needs to be justified. The three econometric models have different assumptions about the underlying data generation process. 1) A more detailed discussion about model selection between OLS, SLM and SEM is necessary. The exclusion of other more general spatial econometric models also needs to be discussed. 2) In addition, in Table 5 and 6, SLM seems to be more appropriate in some periods while SEM seem more appropriate in other periods. This time inconsistency in the underlying data generation process needs to be justified.
Response: Thank you for your suggestion. We have refined the discussion about model selection and rechecked Table 5 and 6 in the revised manuscript.
1) We have quoted a new reference (No. 64) to give reasons why SLM and SEM should be employed other than OLS, these two spatial regression models could describe and explain the related problems arising from spatial effects. Please find them in line 292-295 (all markup mode of the word) in revised subsection 3.2. Besides, standards to judge a superior model are already given in line 315-318 in this subsection. To prove the reasonability, we add a new citation (No. 65). In subsection 4.2, we have verified the standards, so it is clear that the spatial regression models are superior to the OLS, and SEM is better than SLM in most years. Please find them in line 334-340 (all markup mode of the word) in the revised manuscript.
2) Thank you for reminding us to recheck the results in Table 5 and 6. According to the estimation results in the tables, SEM and SLM could both reveal the driving mechanism affecting rural transition and its dynamic change from 1995 to 2015 in the same way, the tiny differences are just between values of regression coefficients of the variables. Inappropriate sentences are changed in subsection 4.3, please find them in line 374-376 (all markup mode of the word) in the revised manuscript.
- The authors describe the merge of Xiangs over time. This boundary change invalidates the reported comparisons over time. One alternative is to construct a time-consistent administrative boundary and then conduct a panel analysis.
Response: Thank you for your suggestion. We have clarify the process of township merging movement in the revised subsection 2.2, a new Figure 1(b) also demonstrates the adjustment of township-level boundaries in 1990 and 2015. We appreciate the idea of panel analysis, however, the purpose of this study is to analyze the dynamic change of driving mechanism in different periods. So the method of cross-section data analysis in this paper is more appropriate to the research purpose. By comparing the regression coefficient of core variables in different years, we have summarized the main driving mechanism of rural transition in Suzhou, such as “non-grain” production trend, spatial spillovers effect, and contribution from new industry.
- In the model specification, the potential endogeneity problem needs to be carefully addressed.
Response: Thank you for your suggestion, we have considered the endogeneity problem in this paper. In the revised subsection 4.1, collinearity diagnostics of indicators are done before next step of statistical test. Please find the results of collinearity diagnosis and explanations in line 320-329 (all markup mode of the word) in the revised manuscript. Similar analysis on the driving mechanism of rural transition can be found in revised subsection 5.2 and 5.3.
Reviewer 2 Report
The manuscript presents an interesting topic. There are a few suggestions for improvement:
1) In the abstract, using the abbreviations of the index (R3, U4 etc) is not very clear since these are not standard abbreviations.
2) In the introduction the gap in the literature to which the research intends to respond should be better emphasized
3) Please provide references for the used methods
3) The findings/discussions should be discussed with respect to other studies in the field.
4) What are the limitations of the research?
5) What about the theoretical and practical implications?
Author Response
Reviewer 2
The manuscript presents an interesting topic. There are a few suggestions for improvement:
- In the abstract, using the abbreviations of the index (R3, U4 etc) is not very clear since these are not standard abbreviations.
Response: Thank you for your suggestion, we have revised the abstract. Abbreviations of the index (e.g., R3) and the model (e.g., SLM) are all refined. We apologize for the mistakes.
- In the introduction the gap in the literature to which the research intends to respond should be better emphasized
Response: Thank you for your suggestion, we have revised the introduction part. To emphasize the significance of this paper comparing with other literatures, we have added the following sentence.
Since existing research focus more on rurality evaluation from a macro-scale (i.e., national and regional), it is necessary to analyze dynamic change of the driving mechanism of rural transition from a relative micro-scale, such as a specific city where urban and rural areas have transitioned from “one-way flow” to “bilateral interaction”, and from “urban bias” to “urban–rural integration”.
Please find it in line 113-118 (all markup mode of the word) in the revised manuscript.
- Please provide references for the used methods.
Response: We try to respond to your suggestion in two sections in the revised manuscript. First, in the introduction part, we have added new references (No. 5-7) about rurality and urbanity measurement in the existing research. Second, in the methods part (subsection 3.2 in the revised manuscript), another two new references (No. 64 and 65) are provided for explaining models used in this study, please find them in line 295 and 317 (all markup mode of the word) in the revised manuscript. Reference 64 tells us that SLM and SEM could be used to describe and explain the related problems arising from spatial effects. Reference 65 gives information to judge a superior model with R-squared value, value of Log likelihood, values of Akaike info criterion and the Schwarz criterion.
- The findings/discussions should be discussed with respect to other studies in the field.
Response: Thank you for your suggestion, we have refined the discussion part with other studies as required. In subsection 5.1 in the revised manuscript, we have related the trends of “non-grain” production to grain security in China after quote references No. 66 and 67, implication of policy-making for protecting “rice bowls” is also offered here. Please find them in line 415-420 (all markup mode of the word). Similar analysis on the driving mechanism of rural transition in the discussion part can be found in revised subsection 5.2 and 5.3.
- What are the limitations of the research?
Response: The limitation of this research is only take Suzhou as the study area. However, reasons are given to explain our consideration. In short, our analysis employing spatial regression models is based on town-level unit, two major problems have limited the data collection, namely, inconsistent indicators in yearbooks of different cities, and incontinuity of town-level data when trace to the earlier years. In this case, rare study area like Suzhou could provide consistent annual statistical data of towns since the 1990s at town level. For future study, we wish to compare Suzhou with other cities experiencing rural transition in China. Please find more details in line 513-527 (all markup mode of the word) in the revised manuscript.
- What about the theoretical and practical implications?
Response: Implications of this study are refined in the last paragraph in the revised manuscript, please find them in line 528-544 (all markup mode of the word). Theoretically, a singular, absolute developmental trajectory of “modernization” does not apply to rural transition in China, because the case study of Suzhou indicates that multifunctional rural transition has provided a comparative advantage that could meet the needs of pluralistic values. Practically, the formulation of policy for sustainable development of rural space should encourage the development of the tertiary industry, considering that the modern service industry, rural tourism are now playing more important roles in rural transition.
Reviewer 3 Report
Dear authors, thanks a lot for providing such an interesting study on rural transitions. However, I think it still needs more preparation and a bit more work before being accepted. Please find my suggestions below.
Title: is a bit too long. Try to make it more concise and attractive.
Abstract
The abstract could be improved. You mentioned rural transition as a core topic but it is unclear why Suzhou is picked as a ‘reasonable’ case. Why do you think Suzhou would be a good case to generate patterns and enough insights for China and even the world? Also, it is unclear what methods and how it is applied.
Please full spell each word and avoid using accoynm if possible, such as SLM.
Also, please remember abstract is for a general audience and most people will only read your abstract. But no one will know what is R3 by reading only read your current abstract. Please try to rewrite your abstract and make it easy to read without needing any background or knowledge of your paper.
Introduction
While writing and referencing, please try to split references to each small point you want to reference rather than all together. For example, in line 37, rather than using [1-4], try to do this: economics[1], sociology [2] and geography [3-4]. The advice is to have a maximum of two references for each sub-point.
I think the rural transition emerged in Japan earlier than the 1990s. This topic is not Euro-centric! You need to include more background and theory on rural transitions in Eastern Asia, especially Japan and China. I think it would be beneficial if you could read and include these latest papers: https://doi.org/10.3390/land10050514; https://www.jstor.org/stable/2645373; https://doi.org/10.1016/j.jrurstud.2018.01.013 ; https://doi.org/10.1016/j.jrurstud.2019.12.002;
Line 72, maybe not only in eastern China, but the whole of China.
Line 77, read this paper (https://doi.org/10.3390/land10050514) and please be a bit more specific in the number of lost villages if possible.
Lines 82-86: please legitimise the reason for selecting Suzhou as the case study. Why it could be a ‘reference for studies of rural transition in other rapidly urbanizing areas across the world’.
Materials and methods
Figure 1. (a) should provide more information about the location of Suzhou. It at least needs to include the whole area of Jiangsu, Zhejiang and Shanghai. In contrast, Figure 1. (b) provides too much information. Readers just want to see which regions are urban and which are rural, or peri-urban. Also, this map shows which year’s layout of Suzhou?
Please define what is rurality and what is urbanity. Many villages in China today have a lot of so-called urban characteristics such as well-paved roads and high-speed broadband. Moreover, please explain why secondary industry and tertiary industry are considered urbanity? Rural tourism has now become an important proportion of rural areas. The classification is a bit problematic. Agricultural and non-agricultural division looks more suitable.
Line 173: ‘xiangs’ are equivalent to the township? If so, please add.
For the index, why U5 is for urbanity?
Discussion
Line 350, the data of figure 2 is a bit too aged.
Land is an international journal and I think it is necessary to have a more in-depth engagement with the latest papers on rural transitions above Jiangsu and distil some insights for the international rural scholarship. Please have a look at the papers I provided before. I attached them here again.
https://doi.org/10.3390/land10050514; https://www.jstor.org/stable/2645373; https://doi.org/10.1016/j.jrurstud.2018.01.013 ; https://doi.org/10.1016/j.jrurstud.2019.12.002;
Minor ones:
Lines 6-14: Your correspondences are the same and please only type once. You may not need to list all authors' emails in the manuscript. Of course, this is up to the editor’s decision.
Lines 17-18: continuum of what? Unclear
Line 35: delete ‘in a series of academic publciations’ …it only adds little to your paper. You could say Many veteran scholars or so
Line 108: what is a flourishing location? Unclear? With fertile land?

Author Response
Reviewer 3
Dear authors, thanks a lot for providing such an interesting study on rural transitions. However, I think it still needs more preparation and a bit more work before being accepted. Please find my suggestions below.
- Title: is a bit too long. Try to make it more concise and attractive.
Response: Thank you for your suggestion, we have revised the title as “Analyzing the driving mechanism of rural transition from the perspective of rural-urban continuum: A case study of Suzhou, China”.
- Abstract
1) The abstract could be improved. You mentioned rural transition as a core topic but it is unclear why Suzhou is picked as a ‘reasonable’ case. Why do you think Suzhou would be a good case to generate patterns and enough insights for China and even the world?
2) Also, it is unclear what methods and how it is applied.
3) Please full spell each word and avoid using accoynm if possible, such as SLM. Also, please remember abstract is for a general audience and most people will only read your abstract. But no one will know what is R3 by reading only read your current abstract. Please try to rewrite your abstract and make it easy to read without needing any background or knowledge of your paper.
Response: We have revised the abstract as required.
1) Please find the following sentence in line 20-22 (all markup mode of the word) in the revised manuscript.
This paper takes Suzhou, a rapidly urbanizing prefecture-level city that has experienced three obvious stages of rural transition since China’s reform and opening-up, as the case area to explore the driving mechanism from the perspective of rural-urban continuum.
We have also refined the explanation why Suzhou is a good case in the introduction part, please find sentences in line 113-121 (all markup mode of the word) in the revised manuscript.
2) Please find the following sentences in line 24-26 (all markup mode of the word) in the revised manuscript.
We first construct the index system for measuring rural transition from two dimensions of rurality and urbanity. Then, we identify the core influencing factors of different phases from 1990 to 2015 employing spatial regression models and then extracts the main driving mechanism.
3) We apologize for the mistakes in the abstract. Abbreviations of the index (e.g., R3) and the model (e.g., SLM) are all refined.
- Introduction
1) While writing and referencing, please try to split references to each small point you want to reference rather than all together. For example, in line 37, rather than using [1-4], try to do this: economics [1], sociology [2] and geography [3-4]. The advice is to have a maximum of two references for each sub-point.
2) I think the rural transition emerged in Japan earlier than the 1990s. This topic is not Euro-centric! You need to include more background and theory on rural transitions in Eastern Asia, especially Japan and China. I think it would be beneficial if you could read and include these latest papers: https://doi.org/10.3390/land10050514; https://www.jstor.org/stable/2645373; https://doi.org/10.1016/j.jrurstud.2018.01.013; https://doi.org/10.1016/j.jrurstud.2019.12.002.
3) Line 72, maybe not only in eastern China, but the whole of China.
4) Line 77, read this paper (https://doi.org/10.3390/land10050514) and please be a bit more specific in the number of lost villages if possible.
5) Lines 82-86: please legitimise the reason for selecting Suzhou as the case study. Why it could be a ‘reference for studies of rural transition in other rapidly urbanizing areas across the world’.
Response: 1) Thank you for your suggestion, we have rechecked the whole manuscript to avoid the situation happen again.
2) We have read the four recommended articles carefully, and have added three of them as new references in the revised manuscript, which are No. 47, 67 and 70. We appreciate the idea that there should be more background and theory on rural transition, especially in Asian countries. As a result, a new subsection is added, please find “2.1. Literature review on rural transition” in the revised manuscript. We have reviewed the transition theory emerged in developed countries and policies put forward to deal with rural transition in Asian countries. Such as the “one village, one product” strategy in Japan, the “Saemaul Undong” program (i.e., New Village Movement) in Korea and rural revitalization strategy in China.
In the original manuscript, what we meant is not “rural transition first emerged in Great Britain in the 1990s”. Since China launched the reform and opening-up policy in the 1980s, an early phase of rural transition called Sunan model is took place in the southern Jiangsu province. So, there is no doubt that rural transition emerged in Japan earlier than the 1990s, because urbanization and industrialization occurred much earlier in Japan than in China. What we want to express is that “academic research on rural transition first emerged in the 1990s”. Many articles about rural transition have discussed this issue. Here we offer a latest one published in Geoforum in 2022 (https://doi.org/10.1016/j.geoforum.2021.12.008), some useful information is as follows.
Since the 1990s, a central debate that framed the discussion on rural change has been the transition from “productivist” to “post-productivist” agricultural and rural spaces in advanced economies (Lowe et al., 1993; Ward, 1993; Wilson, 2001). In this lively debate, however, developing countries have been left out, as this postulated transition was largely an experience restricted to Europe and applying post- productivist theory to a developing world context has been highly problematic (Wilson and Rigg, 2003).
3) Thank you for your suggestion, we have deleted the word “eastern”, please find it in line 104 (all markup mode of the word) in the revised manuscript.
4) Thank you for the recommendation of the article, we quite agree with the saying “rural development is reoriented from productivism to multifunctionality” in this paper and its discussion on new trajectories of rural development (e.g., Figure 2). We have quoted this paper in line 494 in the revised manuscript. Meanwhile, what you mentioned about “some villages have declined or even disappeared” is a trajectory of rural transition we concluded from the process of urbanization. We also list other types of trajectories describing rural transition in this paragraph, such as type of modern agricultural villages, industrial villages, and tourism villages. Our point is that there are multiple trajectories of rural transition in China, however, the specific number of lost villages or any other types of transitional trajectory is far by beyond our ability. Please understand this.
5) Thank you for your suggestion, we have refined the reason why selecting Suzhou as the case study, and saying “across the world” is deleted. Please find the following sentences in line 113-121 (all markup mode of the word) in the revised introduction part.
Since existing research focus more on rurality evaluation from a macro-scale (i.e., national and regional), it is necessary to analyze dynamic change of the driving mechanism of rural transition from a relative micro-scale, such as a specific city where urban and rural areas have transitioned from “one-way flow” to “bilateral interaction”, and from “urban bias” to “urban–rural integration”. Therefore, in this paper, we take Suzhou—a rapidly urbanizing prefecture-level city located in the Yangtze River Delta (YRD) region and has experienced three obvious stages of rural transition since the reform and opening up in the late 1980s—as the study area.
- Materials and methods
1) Figure 1. (a) should provide more information about the location of Suzhou. It at least needs to include the whole area of Jiangsu, Zhejiang and Shanghai. In contrast, Figure 1. (b) provides too much information. Readers just want to see which regions are urban and which are rural, or peri-urban. Also, this map shows which year’s layout of Suzhou?
2) Please define what is rurality and what is urbanity. Many villages in China today have a lot of so-called urban characteristics such as well-paved roads and high-speed broadband. Moreover, please explain why secondary industry and tertiary industry are considered urbanity? Rural tourism has now become an important proportion of rural areas. The classification is a bit problematic. Agricultural and non-agricultural division looks more suitable.
3) Line 173: ‘xiangs’ are equivalent to the township? If so, please add.
4) For the index, why U5 is for urbanity?
Response: 1) Thank you for your suggestion, we have redrawn the Figure 1 as required. After modification, Figure 1(a) is about location of Suzhou in the Yangtze River Delta region, China; Figure 1(b) and (c) are about change of towns and urban-rural areas in Suzhou from 1990 to 2015.
2) Thank you for your suggestions on the definition of rurality and urbanity, and indices to measure them. First, we want to quote what Chung (2013) said in the article “Rural transformation and the persistence of rurality in China” published in Eurasian Geography and Economics (http://dx.doi.org/10.1080/15387216.2014.902751).
China’s large land mass and diversity have made the conceptualization of rurality a difficult task. Although in many parts of China “rurality” still means agriculture, in many villages non-agricultural activities have developed side by side with cultivation. More and more farmers have become involved in non-agricultural production and receive occupational income, but they have not become divorced from agriculture.
In fact, scholars hold different opinions about the definition of rurality and urbanity. Here are several of them from the latest literatures.
…starting in the mid-20th century, social scientists increased efforts to move away from a rural–urban dichotomy towards a continuum ranging from remote rural to dense urban settings with territorial classifications of the relative degree of ‘‘rurality” or ‘‘urbanicity.” (Cattaneo et al., 2022. https://doi.org/10.1016/j.worlddev.2022.105941)
Rural areas include smaller settlements than urban areas, whereby the size of settlements is an important criterion for separating urbanity and rurality. (Balta and Atik, 2022. https://doi.org/10.1016/j.landusepol.2022.106144)
By referencing a case study pertaining to urban-rural planning policy, I have demonstrated that spatial concepts, such as rural and urban, are not fixed and stable entities. Rather, these concepts are subject to modification, thanks to the circulation of knowledge. With a particular emphasis on the shifting notions of rural-urban relationships and rurality, this paper questions the enduring rural-urban dichotomy. Rural-urban interfaces no longer stand for the beginning and end of rurality and urbanity, but accommodate hybrid forms of human settlement. (Wang, 2022. https://doi.org/10.1016/j.jrurstud.2022.03.016)
Another article about gentrification of a village may also answer the question why there are urban characteristics in rural area.
The relevance of these ideas to gentrification is then explored through an investigation of the gentrification of a village, which like many ‘urban’ settlements, has experienced both industrialisation and deindustrialisation. (Phillips et al., 2022. https://doi.org/10.1016/j.jrurstud.2022.02.004)
Enlighten by the above studies and others, we try to define the rural space from a perspective of rural-urban continuum. Rurality means rural characteristics while urbanity is urban characteristics in rural area. So to explain why secondary industry and tertiary industry are considered urbanity, we think both secondary and tertiary industries are result of urbanization and industrialization during the rural transition, it is clear that they stand for urban characteristics in rural area.
3) As we explained in the revised subsection 2.2 “study area”, xiang is another administrative unit similar to town in China. However, as a result of township merging movement during the rapid urbanization, all xiangs in Suzhou were merged into cities or towns by 2015. Please find these sentences in line 190-193 (all markup mode of the word) in the revised manuscript.
4) Basically, we think the index “per capita rural electricity consumption” (U5) could represent the modern life of villagers, which is an urban characteristic in rural area driven by urbanization and industrialization during the rural transition. Here we quote a sentence from Cattaneo et al. (2022) about electricity as modern infrastructure service.
For example, Ferré and co-authors (2012) find that poverty is both more widespread and deeper in very-small and small towns than in large or very large cities, generally due to lack of access to basic infrastructure services, such as electricity.
- Discussion
1) Line 350, the data of figure 2 is a bit too aged.
2) Land is an international journal and I think it is necessary to have a more in-depth engagement with the latest papers on rural transitions above Jiangsu and distil some insights for the international rural scholarship. Please have a look at the papers I provided before.
Response: 1) We have update the data to 2020 in Figure 2.
2) Thank you for your suggestion, we understand that Land is an international journal and some latest papers have been added in the revised discussion part, so as to compare with existing studies and provide more information for international scholars in this field. For example, in the revised subsection 5.1, we have related the trends of “non-grain” production to grain security in China after quote references No. 66 and 67, implication of policy-making for protecting “rice bowls” is also offered. Reference No. 67 is from your recommendation. Please find them in line 415-420 (all markup mode of the word) in the revised manuscript. Similar analysis on the driving mechanism of rural transition in the discussion part can be found in revised subsection 5.2 and 5.3.
- Minor ones:
1) Lines 6-14: Your correspondences are the same and please only type once. You may not need to list all authors' emails in the manuscript. Of course, this is up to the editor’s decision.
2) Lines 17-18: continuum of what? Unclear
3) Line 35: delete ‘in a series of academic publciations’ …it only adds little to your paper. You could say Many veteran scholars or so
4) Line 108: what is a flourishing location? Unclear? With fertile land?
Response: 1) Thank you for your suggestion, we have modified the correspondence information.
2) We have refined it as “rural-urban continuum”.
3) Thank you for your suggestion, the sentence is modified.
4) We have deleted the word in the revised subsection 2.2 “study area”.
Reviewer 4 Report
Dear Authors,
The paper is well written: English spelling and grammar are good.
The manuscript begins with an informative and well-structured abstract.
In the Introduction, in my opinion, the literature that was used is dated although appropriate for the subject of the paper being conducted on academic articles. The objectives of the paper need to be more clearly stated ("to provide a reference for studies of rural transition in other rapidly urbanizing areas across the world" is not enough as an objective), and the outcomes need to be more clearly aligned to the objectives.
The data collection and analysis methods used in the research are clearly explained. The analysis is performed correctly and accurately.
The results are presented in a comprehensive way and appropriate sequence. All data presented in tables and figures are easily understandable.
In the discussion section, the results are almost well discussed, but the use of scientific literature is nil. In the INTRODUCTION, reference is made to rural areas and policies in Europe and North America, but then there is no trace of such a comparison in the discussions, which would make the discussion of the results robust.
The major weakness of this study is its completely missing theoretical and/or conceptual background which is usually required in academic journals (except for the shorter research notes). Also, a detailed literature review section about the rural transition is completely missing (some aspects are included in the Introduction together with section 2.1. Study area; these should be expanded and put into a separate section before "Materials and methods").
The manuscript contains a relatively large number of dated bibliographical references; the bibliography used should be expanded with more recent articles.
The conclusion section lacks limitations of the study and further research to be developed.
Author Response
Reviewer 4
Dear Authors,
The paper is well written: English spelling and grammar are good.
The manuscript begins with an informative and well-structured abstract.
- 1) In the Introduction, in my opinion, the literature that was used is dated although appropriate for the subject of the paper being conducted on academic articles. 2) The objectives of the paper need to be more clearly stated ("to provide a reference for studies of rural transition in other rapidly urbanizing areas across the world" is not enough as an objective), and the outcomes need to be more clearly aligned to the objectives.
Response: Thank you for your suggestion, we have revised the introduction part as required. 1) Since the title of the paper is changed, we have re-written the first paragraph of the paper, and have added some literatures on the subject of rural-urban continuum published in the recent years (references No. 1-9). Please find them in line 67-85 (all markup mode of the word) in the revised manuscript.
2) The objectives of the paper is modified as “analyze dynamic change of the driving mechanism of rural transition from a relative micro-scale”. The last paragraph of the introduction part has indicated the procedures to fulfill the objectives. Specifically, the spatial regression models are employed to measure the town-level unit of Suzhou to sort out key driving factors in different periods. As a result, three main driving mechanism of rural transition are summarized in the revised discussion part. Theoretical and practical implications for policy-making are also offered in the revised conclusions part.
- The data collection and analysis methods used in the research are clearly explained. The analysis is performed correctly and accurately.
The results are presented in a comprehensive way and appropriate sequence. All data presented in tables and figures are easily understandable.
In the discussion section, the results are almost well discussed, but the use of scientific literature is nil. In the INTRODUCTION, reference is made to rural areas and policies in Europe and North America, but then there is no trace of such a comparison in the discussions, which would make the discussion of the results robust.
Response: Thank you for your suggestion, we have added some latest papers in the revised discussion section, so as to compare with existing studies and nourish the results. For example, in the revised subsection 5.1, we have related the trends of “non-grain” production to grain security in China after quote references No. 66 and 67, implication of policy-making for protecting “rice bowls” is also offered after a comparison. Please find them in line 415-420 (all markup mode of the word) in the revised manuscript. Similar analysis on the driving mechanism of rural transition in the discussion section can be found in revised subsection 5.2 and 5.3.
- The major weakness of this study is its completely missing theoretical and/or conceptual background which is usually required in academic journals (except for the shorter research notes). Also, a detailed literature review section about the rural transition is completely missing (some aspects are included in the Introduction together with section 2.1. Study area; these should be expanded and put into a separate section before "Materials and methods").
Response: We have added a new subsection “2.1. Literature review on rural transition” before the “2.2. Study area” as required. To supplement the literature review section about the rural transition, we first reviewed the transition theory development in developed countries, such as productivism, post- productivism and multifunctionality. Then we combed policies put forward to deal with rural transition in Asian countries. Such as the “one village, one product” strategy in Japan, the “Saemaul Undong” program (i.e., New Village Movement) in Korea and rural revitalization strategy in China. Lastly, we summarized the main aspects of studies on agricultural and rural transition by scholars in China.
- The manuscript contains a relatively large number of dated bibliographical references; the bibliography used should be expanded with more recent articles.
Response: Thank you for your suggestion, we have updated the bibliography in the revised manuscript with more than 40 new references, most of them are published in recent years.
- The conclusion section lacks limitations of the study and further research to be developed.
Response: The limitation of this research is only take Suzhou as the study area. However, reasons are given to explain our consideration. In short, our analysis employing spatial regression models is based on town-level unit, two major problems have limited the data collection, namely, inconsistent indicators in yearbooks of different cities, and incontinuity of town-level data when trace to the earlier years. In this case, rare study area like Suzhou could provide consistent annual statistical data of towns since the 1990s at town level. For future study, we wish to compare Suzhou with other cities experiencing rural transition in China. Please find more details in line 513-527 (all markup mode of the word) in the revised manuscript.
Round 2
Reviewer 1 Report
The same concerns in the previous review are not adequately addressed.
For comment (2), since both spatial lag and spatial error model might be valid, a spatial Durbin model should be applied and test statistics on the spatial lag and spatial error models should be reported.
The boundary change invalidates the comparison over time. Even when the towns and Xiangs share the same name, they are not comparable because they cover different spatial regions, populations, and economic structures.
For the endogeneity problem, checking the collinearity is insufficient.
Author Response
- The same concerns in the previous review are not adequately addressed.
For comment (2), since both spatial lag and spatial error model might be valid, a spatial Durbin model should be applied and test statistics on the spatial lag and spatial error models should be reported.
Response: Thank you for your suggestion, we have added the diagnostics for spatial dependence in the revised subsection 4.2. Test reports of LM-Lag and LM-Error are provided, please find the new table 4 in line 301-303 (all markup mode of the word) and relevant explanation in line 289-295 in the revised manuscript.
- The boundary change invalidates the comparison over time. Even when the towns and Xiangs share the same name, they are not comparable because they cover different spatial regions, populations, and economic structures.
Response: Thank you for your comment. It’s true that boundaries of towns and xiangs may change over time, especially after the township merging movement during the rapid urbanization, as we mentioned in subsection 2.2. However, we think this change will not affect the estimation units (towns and xiangs) in every single year because we use the method of cross-section data analysis in this paper. Estimation results of each year are generated from estimation units of the current year, so there is no need to consider the boundary change and comparison of the same town (xiang) in different years.
- For the endogeneity problem, checking the collinearity is insufficient.
Response: Thank you for your suggestion, we are deeply sorry for the inappropriate explanation last time. This problem is highly valued and fully discussed by all the authors. After consulting experts in econometric model field and searching relevant literatures, we try to respond to your concern of endogeneity problem by providing interpretation based on existing literatures.
According to Chen and Wu (2014), reverse causality and omitted variables in model estimation could generate endogeneity problem, and the adoption of instrumental variable could solve the problem.
The panel fixed-effect estimation in the above-mentioned section could be biased due to some reverse causality or omitted variables problem.
To deal with the potential endogeneity problem, we adopt the instrumental variable approach.
Chen, B.R.; Wu, M.Q. Industrial agglomeration and employer compliance with social security contribution: Evidence from China. Journal of Regional Science 2014, 54(4), 586-605.
Further, in the existing literatures (e.g., Au and Henderson, 2006; Barone and Narciso, 2015), instrumental variables are encouraged to choose from the geographical and historical perspective.
It may contain time persistent shocks to do with unmeasured location-geographic features or local political and institutional environments that again affect both productivity and factor allocations.
Our strategy to deal with these problems affecting identification is to instrument for all contemporaneous time-varying covariates with historical characteristics of the city.
Au, C.C.; Henderson, J.V. How migration restrictions limit agglomeration and productivity in China. Journal of Development Economics 2006, 80, 350-388.
Therefore, we instrument current mafia activity with exogenous historical and geographical measures of land productivity.
Barone, G.; Narciso, G. Organized crime and business subsidies: Where does the money go? Journal of Urban Economics 2015, 86, 98-110.
So it is important and necessary to interpret the potential endogeneity problems from the angles of reverse causality and omitted variables. These two aspects have been considered when constructing the index system, especially they are chosen from the geographical and historical perspective.
For the potential problem of omitted variables, we chose the 12 indices from two dimensions of rurality and urbanity as a result of the perspective of rural-urban continuum, to reflect both agricultural and non-agricultural activities, and agricultural and non-agricultural production in rural areas. Indices used to measure rurality could represent the quality of resource endowment, various inputs and outputs of agricultural industry; while indices used to measure the urbanity could determine the level of modernization and urbanization as well as the economic conditions of villagers. Given the relatively comprehensive coverage of the 12 indices in this paper, we think they will cause little impact on the potential problem of omitted variables. For the potential problem of reverse causality, many existing literatures have proved the causality among indices used in this paper, especially from the geographical or historical perspective. For example, indices R1 and R3 are about agricultural activities, causal links between them and the dependent variable (economic development in this paper) have been verified in references (No. 15 and No. 75) in the revised manuscript. Similarly, indices U4 and U5 are about non-agricultural activities in rural area, causal links are verified in in references (No. 57 and No. 3).
The formation and development of rural settlements were rooted in agrarian society, whose basis was the natural habitat and whose backdrop was the agricultural economy.
Ref No.15: Li, H.B.; Yuan, Y.; Zhang, X.L.; et al. Evolution and transformation mechanism of the spatial structure of rural settlements from the perspective of long-term economic and social change: A case study of the Sunan region, China. J. Rural Stud 2022, 93, 234-243.
Thailand remains a major producer and exporter of agricultural products, … the contribution of agriculture to the growth of Thailand's domestic product has not changed much over the past two decades.
Ref No.75: Faysse, N.; Aguilhon, L.; Phiboon, K.; et al. Mainly farming … but what's next? The future of irrigated farms in Thailand. J. Rural Stud 2020, 73, 68–76.
From the late 1970s to the 1980s, while private firms were still prohibited, TVEs in Wuxi captured opportunities provided by economic reforms and became the dominant sector in rural economies.
Ref No.57: Yuan, F.; Wei, D.Y.H.; Chen, W. Economic transition, industrial location and corporate networks: Remaking the Sunan Model in Wuxi City, China. Habitat Int 2014, 42, 58-68.
... poverty is both more widespread and deeper in very-small and small towns than in large or very large cities, generally due to lack of access to basic infrastructure services, such as electricity.
Ref No.3: Cattaneo, A.; Adukia, A.; Brown, D.L.; et al. Economic and social development along the urban–rural continuum: New opportunities to inform policy. World Development 2022, 157, 105941.
Please find the detailed explanation on the potential endogeneity problem in line 335-350 (all markup mode of the word) in the revised manuscript.
Reviewer 2 Report
The manuscript was improved based on recommendations and from my point it is ready for publication.
Author Response
The manuscript was improved based on recommendations and from my point it is ready for publication.
Response: Thank you very much for your decision. Please accept our gratitude for all your suggestions and comments during the revision process, they are very important and valuable to the paper.
Reviewer 3 Report
Dear authors,
Thanks for preparing the revised version of your paper and addressing my comments point-by-point. This manuscript has been much improved.
There are a few things you may need to consider addressing before accepting.
lines 127-129. China's rural development/revitalisation policy is a milestone and there is a lot in this policy. But you just described the five principles but did not explain the significance of the policy for rural development and transition. I suggest using the references I provided last time and adding a few insights here.
Lines 179-180: who believes this? unclear
Lines 210-211: I think I request a definition of the rurality and urbanity but they are not shown here. Please define them. Also, please explain why you select the 12 indices.
Lines 345-346: should add a few ecological impacts of pond fish farming, such as the biological escape of invasive species to the natural body and also nutrient runoffs
Author Response
Dear authors,
Thanks for preparing the revised version of your paper and addressing my comments point-by-point. This manuscript has been much improved.
There are a few things you may need to consider addressing before accepting.
- lines 127-129. China's rural development/revitalisation policy is a milestone and there is a lot in this policy. But you just described the five principles but did not explain the significance of the policy for rural development and transition. I suggest using the references I provided last time and adding a few insights here.
Response: Thank you for your suggestion, explanation on the significance of the rural revitalization strategy plan is added in the revised subsection 2.1. We think the significance for rural transition is aim at solving key problems relating to rural transition, and villages are encouraged to pursue appropriate transition pathways according to the policy, so to become revitalized in the long run. Please find this expression in line 128-134 (all markup mode of the word) in the revised manuscript.
- Lines 179-180: who believes this? Unclear
Response: Thank you for your comment, according to the reference we have cited here (No. 5), people who believe this are researchers of continuous studies about the evaluation of rurality. Please find the sentences in the reference we cited.
Given this, continuous studies about the evaluation of rurality have been carried out in various countries and regions like England and Wales (Cloke, 1977, Cloke and Edwards, 1986, Harrington and O'Donoghue, 1998), Spain (Ocaña-Riola and Sánchez-Cantalejo, 2005, Prieto-Lara and Ocaña-Riola, 2010), USA(Waldorf, 2006), Turkey (Gülümser et al., 2008, Gülümser et al., 2009), Serbia (Bogdanov et al., 2008), Nigeria (Madu, 2010), China (Long et al., 2009a, Meng et al., 2013), and so on. It is believed that the index provides a useful tool which is able to give an insight not only into the static distribution of rurality, but also into the processes of rural change over time.
To make the sentence clearer, we have revised it as “…, who believed that…” Please find it in line 183 (all markup mode of the word) in the revised manuscript.
- Lines 210-211: I think I request a definition of the rurality and urbanity but they are not shown here. Please define them. Also, please explain why you select the 12 indices.
Response: Thank you for your suggestion, a definition of the rurality and urbanity is added in line 196-205 (all markup mode of the word) in the revised manuscript, some articles are also quoted to support the definition.
As mentioned in the original manuscript, 12 indices are selected from two dimensions of rurality and urbanity as a result of the perspective of rural-urban continuum. Indices used to measure rurality could represent the quality of resource endowment, various inputs and outputs of agricultural industry; while indices used to measure the urbanity could determine the level of modernization and urbanization as well as the economic conditions of villagers. However, the principle to make this choice needs more explanation. We think the establishment of index system should be combined with previous studies in this paper, which means the reflection of both agricultural and non-agricultural activities, and agricultural and non-agricultural production in rural areas is important. Please find the revised part in line 216-219 and line 228-229 (all markup mode of the word).
- Lines 345-346: should add a few ecological impacts of pond fish farming, such as the biological escape of invasive species to the natural body and also nutrient runoffs
Response: Thank you for your suggestion, relevant information is added as required. Considering that we are not familiar with the ecological impacts, we only mentioned potential risks of invasive fish and eutrophication in the paper, please find the new sentence in line 389-390 (all markup mode of the word). Pond fish farming industry in this paper stands for non-grain production on cultivated land, which could threat grain security if the rapid expansion situation continues in rural areas. So in the original manuscript, we only emphasize the problem from the perspective of grain security in China, not from the angle of ecological impacts.
Reviewer 4 Report
Dear authors,
well done! The manuscript has been much improved. Thank you for addressing my comments point-by-point.
Best wishes
Author Response
Dear authors,
well done! The manuscript has been much improved. Thank you for addressing my comments point-by-point.
Best wishes
Response: Thank you very much for your decision. Please accept our gratitude for all your suggestions and comments during the revision process, and best wishes to you too.